# Physico-Mechanical Properties of Waterlogged Archaeological Wood: The Case of a Charred Medieval Shipwreck

**Eirini Mitsi** [1] , **Nikolaos-Alexios Stefanis** [2] **and Anastasia Pournou** [2,*]

1 Department of History, Archaeology, and Cultural Resources Management, University of the Peloponnese, 24100 Kalamata, Greece; eir.mitsi@gmail.com
2 Department of Conservation of Antiquities and Works of Art, University of West Attica, Ag. Spyridonos Str., Aegaleo, 12243 Athens, Greece
* Correspondence: pournoua@uniwa.gr

**Abstract:** In 2008, a late-12th-century merchant ship was discovered off the commercial port of Rhodes. The vessel caught fire before sinking and thus numerous hull timbers were found charred. Three main degrees of charring have been recorded that presented major chemical differences which indicated different conservation requirements. This study investigated the correlation between the chemistry of the waterlogged timbers and their physico-mechanical properties, to assist in the development of an appropriate conservation strategy. Scanning electron microscopy documented the morphology of charred, semi-charred and uncharred samples. Moisture content and density were measured gravimetrically, while porosity was evaluated using mercury intrusion porosimetry. Hardness was assessed using a modified Janka test and a penetrometer. The results obtained showed that differences in chemistry were highly correlated to the physico-mechanical properties of the timbers. The charred wood presented the lowest moisture content, shrinkage and porosity among the three charring conditions and it also had the highest density, Janka hardness and resistance to penetration. The exact reverse properties were recorded for the uncharred material, which was typical of badly preserved, waterlogged wood. The semi-charred wood presented transitional features. These results indicate that the uncharred wood is in need of consolidation, in contrast to the charred and semi-charred material, which may be left to air-dry untreated.

**Keywords:** waterlogged wood; charred wood; pyrolysis; thermal degradation; physical properties; porosimetry; Janka hardness; SEM



## 1. Introduction

In 2008, during an underwater survey conducted by the Greek Ephorate of Underwater Antiquities, a late-12th-century merchant ship was discovered off the commercial port of Rhodes, at a depth of ~14 m [1]. Excavation of the shipwreck revealed that the vessel caught fire before sinking, as evidence of burning was recorded on almost every recovered artifact and on numerous construction elements of the ship [1,2].

The excavated elements of the wooden hull presented a varied state of preservation [3,4], as the fire affected differently the degree and depth of wood charring. Thus, coexistence of uncharred, semi-charred and charred wood was often encountered even on the same timber element [3].

This inhomogeneous charring of the wreck is due to several parameters including fire factors, such as the heat flux, temperature and duration [5–11]; the ambient parameters surrounding the wood [5,8,9,12]; and wood variables, such as density, moisture content, permeability, species, size, grain direction and surface protection [5,8,13–15], which also influence the charring rate and depth.

Chemical analysis conducted by Mitsi et al. [4] on the main three charring conditions of the timbers, demonstrated that the charred areas are chemically similar to charcoals, semi-charred parts to thermally modified wood and the uncharred material to biodeteriorated

waterlogged wood. Furthermore, Mitsi et al. [4] showed that in charred areas opposed to uncharred ones, elements such as sulfur, iron and oxygen decrease, while carbon increases.

This dissimilar chemistry of the material is principally due to differing exposure to the fire, which, in addition to the chemistry, is likely to have affected the structure and properties of the wood [7,16–18]. Therefore, it is anticipated that the three main charring conditions identified on the Rhodes shipwreck will also present dissimilar physical and mechanical properties and consequently will have quite different conservation requirements.

This study was set up to investigate the physico-mechanical properties of the Rhodes shipwreck timbers in order to further assess their degree of degradation and contribute to the development of a successful conservation strategy.

## 2. Materials and Methods

The waterlogged wood examined in this study came from one frame of the Rhodes wreck, which was made of *Pinus halepensis* Mill. (Aleppo pine) or *Pinus brutia* Ten. (Turkish pine) [1]. The outer layer of the frame was charred, the inner core was uncharred and in between there was a semi-charred zone. The part of the frame used (Figure 1) measured ~9 cm Ø × 15 cm length. It was retrieved in 2013 and since then it had been kept waterlogged at 5 °C. Samples produced corresponded to sapwood and represented the three main charring conditions.

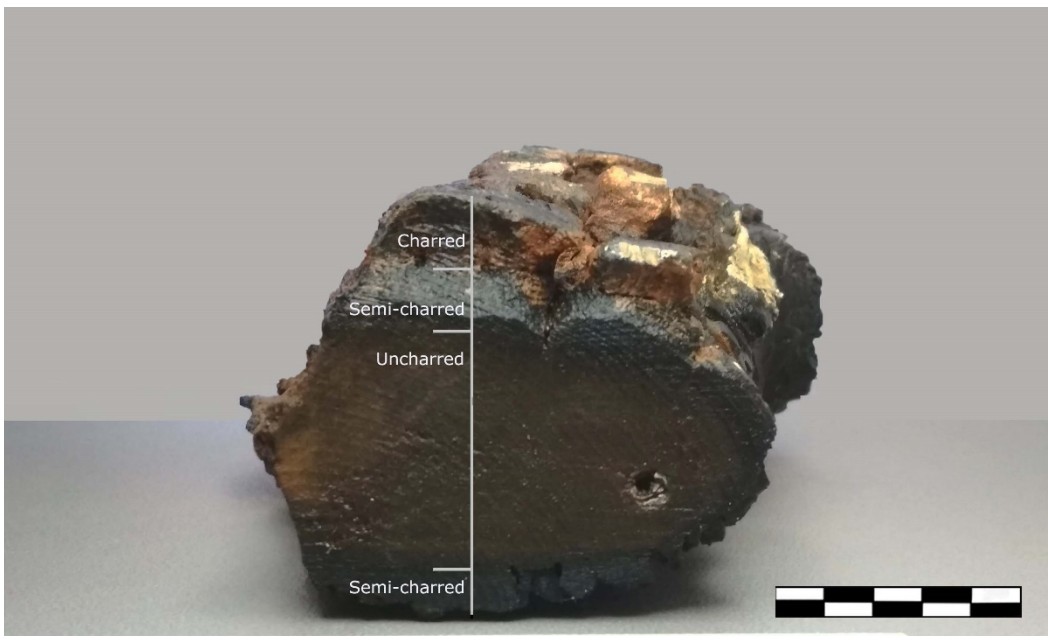

**Figure 1.** The frame part used, presenting an outer charred layer, an uncharred inner core and a semi-charred zone in between.

### 2.1. Morphological Alterations at a Cellular Level
Scanning Electron Microscopy (SEM)

Uncharred and semi-charred waterlogged wood was cut into samples measuring ~0.2 cm × 0.2 cm × 0.3 cm using a double-edged razor blade, whereas charred wood was fractured to produce samples of the same dimensions. Four samples per plane of each charring condition were investigated. All samples were dehydrated in a series of ethanol solutions of increasing concentrations until water-free alcohol was acquired and then left to air-dry in a desiccator. They were then mounted on aluminum stubs using a double-sided carbon conductive tape, gold-coated in a sputter coater (Polaron SC7640) and examined at an acceleration voltage of 20 kV under low vacuum (33 Pa) using a JEOL JSM-6510LV scanning electron microscope.

## 2.2. Physical Properties

2.2.1. Moisture Content, Density and Shrinkage Determination

Waterlogged samples, ~2 cm × 2 cm × 2 cm (tangential (T) × radial (R) × longitudinal (L)), of uncharred, semi-charred and charred wood were used for the determination of moisture content, density and shrinkage.

For the moisture content (MC), samples were weighed in their waterlogged state and oven-dried at 103 ± 2 °C to constant weight [19]. The MC was calculated based on Equation (1) [20].

For the equilibrium moisture content (EMC), the waterlogged samples were air-dried at 21 ± 2 °C and 45 ± 5% RH until they reached equilibrium. The EMC was then calculated based on Equation (2) [21].

The basic density (Rg) of the material was calculated according to Equation (3) [21], based on its constant oven-dry weight, at 103 ± 2 °C, and its waterlogged volume, recorded by water displacement [19]. Relative density (rRg) was estimated based on the weight in air and in water of the waterlogged material according to Equation (4) [22].

For measuring shrinkage, stainless steel insect pins were placed on the transverse plane to mark the tangential and radial direction, and the distance between the pins was recorded with a Vernier caliper (0.02 mm). Samples were then air-dried to constant weight at 21 ± 2 °C and 45 ± 5% RH, and the distance between the pins was measured again. The cross-sectional shrinkage (ß $_{cross}$) was estimated by summing the tangential and radial shrinkage (ß), which had been calculated with Equation (5) [23]. It has to be noted that shrinkage in this study refers to the total reduction of dimensions measured upon air-drying, which includes both cell collapse and cell wall shrinkage.

For evaluating the above-mentioned physical properties, four replicates of each charring condition were used.

$$MC\ (\%) = [(W_W - W_{OD})/W_{OD}] \times 100 \tag{1}$$

where MC = moisture content, $W_W$ = waterlogged weight and $W_{OD}$ = oven-dry weight at 102 ± 3 °C after three consecutive constant measurements.

$$EMC\ (\%) = [(W_{AD} - W_{OD})/W_{OD}] \times 100 \tag{2}$$

where EMC = equilibrium moisture content, $W_{AD}$ = air-dried weight at 21 ± 2 °C and 45 ± 5% RH after three consecutive constant measurements and $W_{OD}$ = oven-dry weight at 102 ± 3 °C after three consecutive constant measurements.

$$Rg\ (g/cm^3) = W_{OD}/V \tag{3}$$

where Rg = basic density, $W_{OD}$ = oven-dry weight at 102 ± 3 °C after three consecutive constant measurements and V = waterlogged volume measured by water displacement.

$$rRg = 3 \times W_{sub}/(W_{air} - W_{sub}) \tag{4}$$

where rRg = relative density, $W_{sub}$ = waterlogged weight in water and $W_{air}$ = waterlogged weight in air.

$$ß\ (\%) = [(l_W - l_{AD})/l_W] \times 100 \tag{5}$$

where ß = shrinkage, $l_W$ = waterlogged distance between pins and $l_{AD}$ = air-dried distance.

Shrinkage was calculated for the tangential (ßt), and for the radial (ßr) direction; for the cross-sectional shrinkage (ß$_{cross}$), the sum, ßt + ßr, was calculated.

2.2.2. Mercury Intrusion Porosimetry (MIP)

Mercury intrusion porosimetry (MIP) was carried out with a Quantachrome PoreMaster 60 on uncharred, semi-charred and charred samples. The archaeological samples of the three charring conditions, measuring approximately 0.5 cm × 0.5 cm × 0.5 cm (T × R × L),

were freeze-dried prior to porosity measurements. This was decided to maintain as much as possible the "original" waterlogged pore structure, as air drying would greatly affect the wood porosity due to collapse and shrinkage. Reference samples of *P. halepensis* and *P. brutia* of the same dimensions were also examined. All samples were stored in a desiccator prior to porosity measurements. Low pressure was first applied at 50 MPa and then samples were placed in the high-pressure station, where pressure up to 400 MPa was employed in equilibration time of 10 s. Surface tension and mercury wetting angle were set at 0.485 Nm$^{-1}$ and 140°, respectively.

### 2.3. Mechanical Properties

#### 2.3.1. Janka Hardness Test

The methods commonly used for assessing the "surface" hardness (< 1 cm) of wood and especially of thermally-modified wood, are the "Brinell" and the "Janka", which are force- and depth-controlled ball tests, respectively [24–30]. The Brinell test was found inappropriate for the material under investigation as the same force could not produce measurable indentations for all three charring conditions. Therefore, a Janka test was adopted in order to investigate differences in the "surface" hardness for uncharred, semi-charred and charred material.

The Janka hardness test was implemented based on the ASTM D143 [31] and the ASTM D1037 [32], with a modification on the ball diameter, using a 2.5 mm ball instead of the standard 11.3 mm, due to the limited size of the available archaeological material. The test was conducted on an Instron 3367 dual-column universal testing machine, with a 2 kN load cell and compression platens of 100 kN maximum load on upper and lower connections. On the upper platen, the 2.5 mm ball was adjusted using a cylindrical neodymium magnet (outer Ø: 0.8 mm, inner Ø: 1.9 mm) of 0.6 mm length. The test speed was set at 0.1 mm/s rate and a 0.62 mm extension was selected to achieve a final ball penetration into the specimen equal to ~$\frac{1}{4}$ of the ball diameter (D). The load-extension data were recorded per 0.1 s with Bluehill 3 software.

A hardness test was conducted for all three charring conditions and on reference samples of sound *P. halepensis* and *P. brutia* for comparison. Measurements were conducted on end grain and at both radial and tangential planes. The use of the 2.5 mm ball on the sound reference samples allowed documentation of differences between earlywood and latewood, and thus, two replicate measurements were taken per growing period and their average was calculated for each plane. Two samples from each charring condition were examined. Uncharred and charred samples measured ~2 cm × 2 cm × 2 cm (T × R × L), whereas the semi-charred measured ~2 cm × 1.5 cm × 2 cm (T × R × L). For every condition, one sample was tested at the waterlogged state to acquire knowledge on the handling of the material before and during conservation and a second one at the freeze-dried state to assess the residual hardness of the material without bias by the water presence. Earlywood and latewood were rarely distinguishable in the archaeological material; thus, two replicate measurements were taken per plane.

Hardness is expressed as the ratio of the applied force to the projected area of contact and it was calculated based on Equation (6) [33]. The projected contact area is treated as the area of a circle created by the ball indenter at the wood surface, and as is typical in the Janka test, the ball is pressed into half of its diameter [31,34]; the radius of the projected area (r) equals the radius of the indenter (R). In the modified Janka test used, the ball was pushed to a depth of $\frac{1}{4}$ of the ball's diameter (Figure 2a), because beyond this limit many of the charred wood samples cracked or/and failed. As the r value is lower than ball radius (Figure 2b), the radius of the projected contact area was geometrically calculated based on the scheme of Figure 2b and Pythagoras' theorem.

$$\text{Hardness} = F/\pi r^2 \tag{6}$$

where F = the force recorded at 0.62 extension, $\pi$ = the mathematical constant (~3.14159) and r = the radius of the projected contact area.

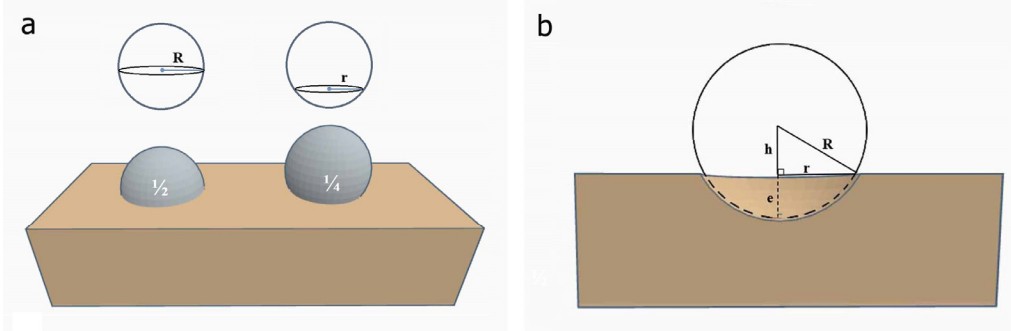

**Figure 2.** (**a**) Graphic representation of the ball penetrated into a depth of $\frac{1}{2}$ and $\frac{1}{4}$ of its diameter, and (**b**) the radius (r) of projected contact area that based on Pythagoras' theorem is equal to $r = \sqrt{(R^2 - h^2)}$, where R = ball radius, h = R − e, and e = extension recorded by Instron.

### 2.3.2. Penetrometer

For assessing the state of preservation of waterlogged archaeological wood, conservators often use the "pin test" [35–39]. This type of manual evaluation is rather subjective; therefore, wood resistance to penetration can be measured with penetrometers instead [40]. These instruments using a minimally invasive procedure can in situ assess the hardness of a material at a given depth and can also indirectly assess its density. In this work, a penetrometer was used for assessing differences in resistance to penetration of the three charring conditions.

Resistance to penetration was recorded on the part of the shipwreck frame (~9 cm Ø, 15 cm length) which included all three charring conditions. A Fruit Hardness Tester, FR- 5105, with a maximum load capacity up to 5000 g was used. The penetrometer was equipped with a needle of 3 cm length and 0.75 mm diameter, of which 1 cm was fasten inside a custom-made holder in order to allow penetration at a constant depth of 2 cm. Six measurements per condition were recorded using the "peak hold" mode on the transverse section of the part, as this was the only section where all charring zones were visible, accessible and could allow discrete penetration into each zone at the same depth without bias.

## 3. Results and Discussion

### 3.1. Morphological Alterations at a Cellular Level

Scanning Electron Microscopy (SEM)

The morphology of the archaeological wood observed with SEM demonstrated the existence of three distinctly dissimilar materials, the uncharred (Figure 3a–c), the semi-charred (Figure 3d–f) and the charred wood (Figure 3g–i). Uncharred waterlogged wood appeared to be severely deteriorated as cells were deformed, while their secondary wall layer presented a granular texture and was commonly detached from the middle lamellae (Figure 3a). In addition, in both tangential (Figure 3b) and radial (Figure 3c) sections, extensive biodeterioration was documented, caused by marine fungi and bacteria, based on the recorded decay patterns [41].

The semi-charred wood was found to be rather intact. Cells appeared to retain their structural integrity and no deformation in their general anatomy was observed (Figure 3d). Cell wall layers were discrete and detachment fissures alongside the middle lamella were rarely detected (Figure 3d). Occasionally, and mostly in longitudinal sections patterns attributed to bacterial and fungal decay were detected (Figure 3e,f).

The differences in the degree of deterioration between uncharred and semi-charred wood are probably because 'thermal modification' of the latter rendered the material less prone to biodeterioration by soft rotters [42,43].

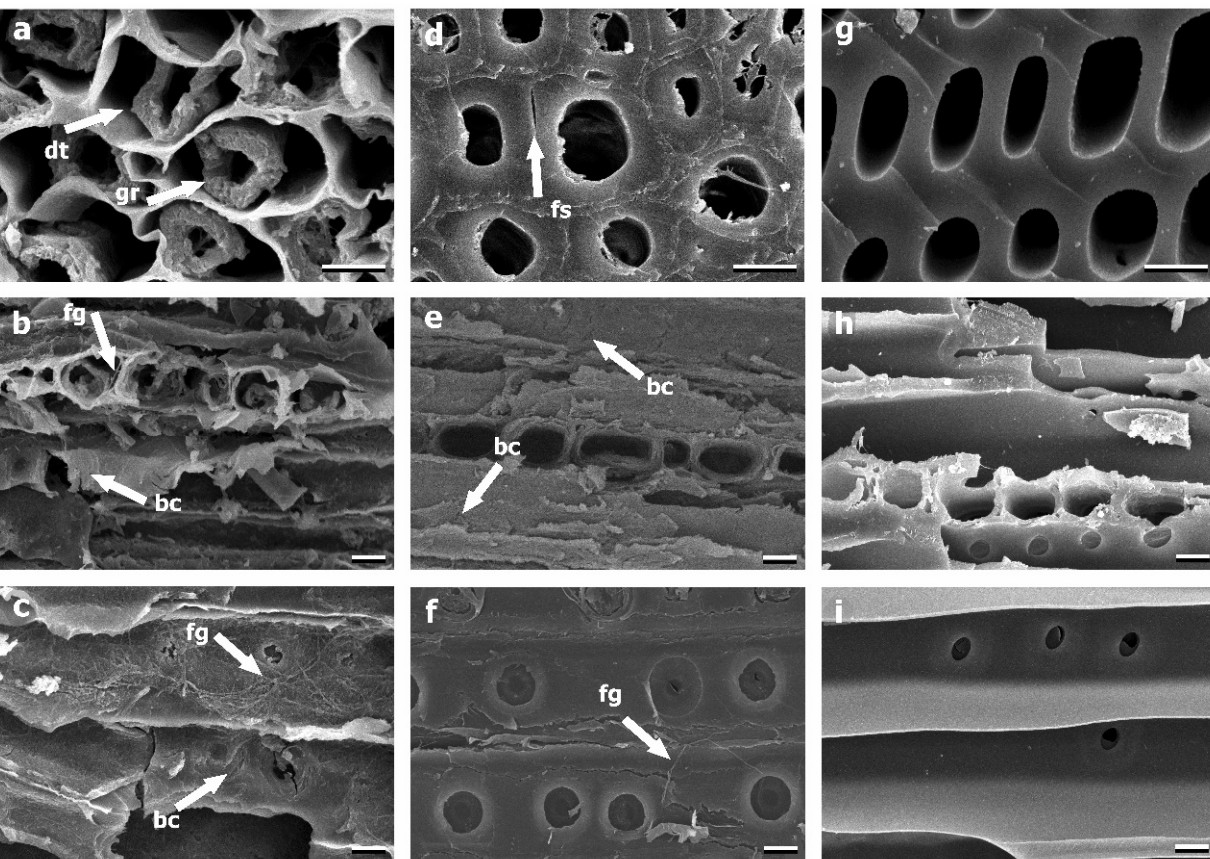

**Figure 3.** SEM micrographs of uncharred, semi-charred and charred wood. Uncharred material (**a**–**c**) presented a granular texture (gr) of the secondary cell walls, which were often detached (dt) from the middle lamellae; semi-charred wood (**d**–**f**) showed intact cell walls with rare detachment fissures (fs); and charred wood (**g**–**i**) presented a vitreous appearance without evident biodeterioration signs. On both uncharred (**b**,**c**) and semi-charred (**e**,**f**) material, biodeterioration patterns caused by bacteria (bc) and fungi (fg) were recognized. All bars are 10 μm.

Lastly, the morphological features observed in charred wood differed greatly from both the uncharred and the semi-charred material. The wood appeared "vitreous" with no signs of deterioration caused by fungi or bacteria (Figure 3g–i). The latter was anticipated, as due to charring most of the organic part of the material had been depleted and thus during burial very few organisms could utilize it. In the transverse section (Figure 3g), cells were slightly distorted possibly indicating fast combustion of wet wood at high temperatures [44]. The middle lamellae appeared to coalesce with the secondary cell walls, a feature that is commonly reported in charred wood [44–50]. This coalescence, also referred to as amalgamation, has been reported to initiate at approximately 300 °C [46,51]. The amalgamation temperature of the material investigated cannot be defined with certainty as it depends on several factors [45–47], nonetheless, it can be safely stated that the charred wood examined has been subjected to temperature ≥ 300 °C. Furthermore, as this feature was absent in the semi-charred material, it could be hypothesized that the exposure temperature was <300 °C, an assumption which is in accordance with its chemical profile studied by Mitsi et al. [4].

### 3.2. Physical Properties

3.2.1. Moisture Content, Density and Shrinkage Determination

Results for the physical properties of the archaeological material are presented in Figure 4 and Table 1 along with reference values of sound *P. brutia* and *P. halepensis*.

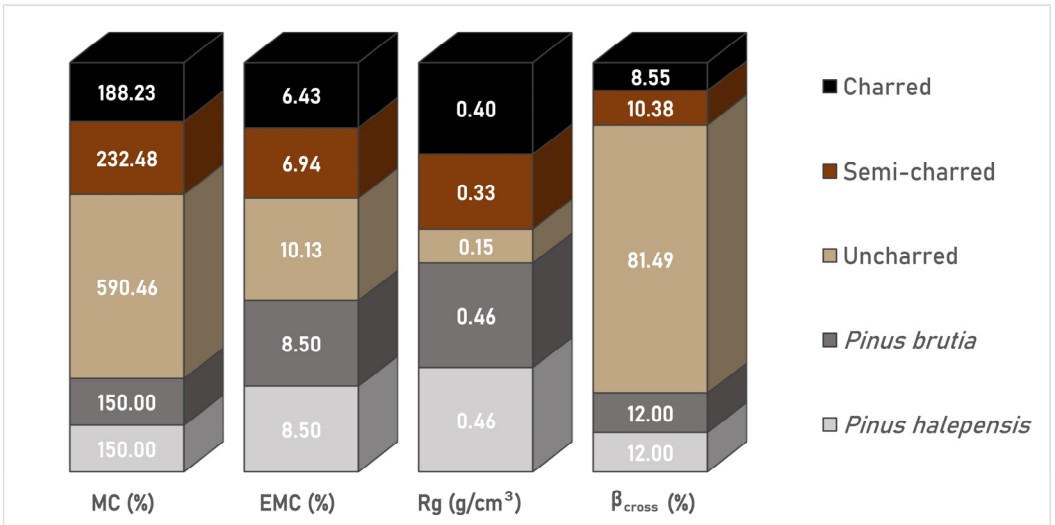

**Figure 4.** Moisture content (MC), equilibrium moisture content (EMC), basic density (Rg) and cross shrinkage ($\beta_{cross}$) values recorded for uncharred, semi-charred and charred archaeological material, juxtaposed to controls of *P. brutia* and *P. halepensis*.

**Table 1.** Moisture content (MC), equilibrium moisture content (EMC), relative density (rRg), basic density (Rg) and cross shrinkage ($\beta_{cross}$) recorded on the archaeological wood and the reference sound wood of *P. brutia* and *P. halepensis*. Values for the archaeological samples are the averages of four replicates. Values in brackets represent the standard deviation.

| Samples | MC (%) | EMC (%) | rRg | Rg (g/cm$^3$) | $\beta_{cross}$ (%) |
|---|---|---|---|---|---|
| Uncharred | 590.46 ($\pm$28.33) | 10.13 ($\pm$0.58) | 0.19 ($\pm$0.02) | 0.15 ($\pm$0.01) | 81.49 ($\pm$0.15) |
| Semi-charred | 232.48 ($\pm$20.58) | 6.94 ($\pm$1.88) | 0.31 ($\pm$0.02) | 0.33 ($\pm$0.02) | 10.38 ($\pm$0.33) |
| Charred | 188.23 ($\pm$24.80) | 6.43 ($\pm$0.47) | 0.38 ($\pm$0.02) | 0.40 ($\pm$0.04) | 8.55 ($\pm$0.40) |
| *Pinus brutia* | 150.00 [a] | 8.00–9.00 [b] | - | 0.46 [c] | 12.00 [d] |
| *Pinus halepensis* | 150.00 [a] | 8.00–9.00 [b] | - | 0.46 [c] | 12.00 [d] |

[a] Umax = [(1/Rg)$-$0.67] $\times$ 100 [23]; [b] [52]; [c] [53]; [d] [54].

The MC of uncharred, semi-charred and charred wood (Table 1) confirmed the waterlogged nature of the material. Furthermore, the values of uncharred wood (590%) were indicative of a highly degraded material [37,55–57]. The values of the semi-charred (232.48%) and the charred wood (188.23%) were lower in comparison to the uncharred material, suggesting that their different exposure to heat had influenced their water holding capacity. More specifically for the semi charred material, the lower MC may partially be attributed to the reduction of hemicelluloses caused by its thermal degradation [4], as the hygroscopicity of wet wood subjected to heat-treatment may be irreversibly reduced [58–61]. Similarly, for the charred material the reduced moisture uptake could be also attributed to the physicochemical alterations caused by the pyrolytic process [62–65].

The equilibrium moisture content (EMC) of the uncharred archaeological wood, at 45 $\pm$ 5% RH, was slightly higher compared to the reference samples. In contrast the EMC of both the semi-charred and charred wood was lower than reference samples (Figure 4). High EMC values have been frequently reported for degraded waterlogged archaeological wood [66–71]. This can be justified by the increased cell wall porosity caused by the action of microorganisms, which increases the bound water of wood [68–70] and consequently the measured values of the "fiber saturation point" (FSP) of waterlogged wood [70]. Furthermore, the small cellulose crystallite length of the material [4] could also be related to a greater availability of sorption sites and thus to the higher EMC values [68,69].

For the semi-charred and charred wood, the reduction of EMC recorded is in line with the reduced water holding capacity of wood exposed to temperatures up to 300 °C [17,72–79]

and above 300 °C [75]. This could be due to the depletion of hemicelluloses, either by biodeterioration in the marine environment [41] or by the heat exposure [17,80], which is negatively correlated to wood EMC as a function of both temperature and duration [72–74,77,81,82]. Nonetheless, it has been shown in heat-treated wood that the EMC reduction is owed to additional mechanisms in addition to the reduced accessibility of OH groups in the cell wall matrix [59,60,83].

Basic density (Rg) values (Table 1) were found to be very low for the uncharred waterlogged samples due to degradation in the marine environment [20,57,66,84]. As anticipated, Rg was also negatively correlated to the MC [36,57,70]. The Rg values of the semi-charred and charred samples were also lower than the references, which is considered to be due principally to the thermal decomposition of the materials at elevated temperatures [27,73,85–88] and to a lesser extent to biodeterioration.

The non-destructive determination of relative density (rRg) gave similar results to the basic density (Rg) for all three charring conditions, and demonstrated that it can be successfully adopted in cases where the oven-dried weight cannot be measured.

Cross shrinkage ($\beta_{cross}$) values for the uncharred waterlogged wood were extremely high (Figure 4), as expected, and negatively correlated with basic density [20,66,84,89]. The same correlation was also recorded for the semi-charred and charred material (Figure 4). However, their shrinkage values were found to be much lower than the uncharred wood values, and to be even lower than the sound reference wood values. This dimensional behavior of the semi-charred wood is more likely to be associated with the low MC of the material that has resulted from thermal degradation.

The results on shrinkage indicate that the semi-charred and charred areas of the wreck are dimensionally stable upon drying, an outcome that should be seriously considered in the conservation strategy of the shipwreck timbers.

### 3.2.2. Mercury Intrusion Porosimetry (MIP)

The pore size distribution recorded according to the IUPAC classification [90] indicates two size classes of pores: the macropores with diameter > 50 nm, (r > 25 nm) and the mesopores with diameters from 2 nm to 50 nm (1 nm < r > 25 nm) (Figure 5). Nonetheless, in wood science the pore size distribution is often categorized in relation to wood structure [91–94]. Therefore, for the softwoods investigated, an anatomy-based categorization has been adopted with three classes (Figure 5): the macrovoids that include the lumen of tracheids and of resin canals with radii ranging from 5 μm to ~200 μm; the microvoids that encompass pit apertures, pit chambers and other small voids with radii from 5 nm to ~5 μm; and the nanovoids that comprise the cell wall porosity with radii < 5 nm [91,95].

Based on the obtained histograms, it is apparent that the pore size distribution among the archaeological wood and the reference *P. brutia* and *P. halepensis* differs considerably (Figure 5). The porosity of reference samples is mainly represented by microvoids, and to a much lesser extent by macrovoids and nanovoids. In contrast, in archaeological wood macrovoids prevailed noticeably as the porosity showed to be shifted towards pores > 1 μm (dashed line). This shifting of the archaeological wood porosity was to be anticipated, as both charring [49,96,97] and biodeterioration in the marine environment may increase wood porosity and permeability [20,37,57,68,70,98].

More specifically, the increased porosity in charred wood is considered to have occurred due to pyrolytic process [49,96,97], where the main volume of macrovoids with radii > 5 μm was created (Figure 5).

Similarly, in the semi-charred material the porosity increase may also be due to the heat exposure, as upon thermal modification the porosity augments as a function of temperature [99,100]. Nonetheless, as most of the organic moieties of the semi-charred wood were shown to be preserved [4], the material was still susceptible to biodeterioration by microorganisms, and it is very likely that its porosity was further increased during burial. Information on thermally modified wood performance against marine fungi and bacteria is scarce [17,101]; however, there are some studies demonstrating that thermal



modification could make wood more resistant to terrestrial soft rot fungi [42,43]. This could probably explain why the porosity of the semi-charred wood was lower compared to the uncharred material.

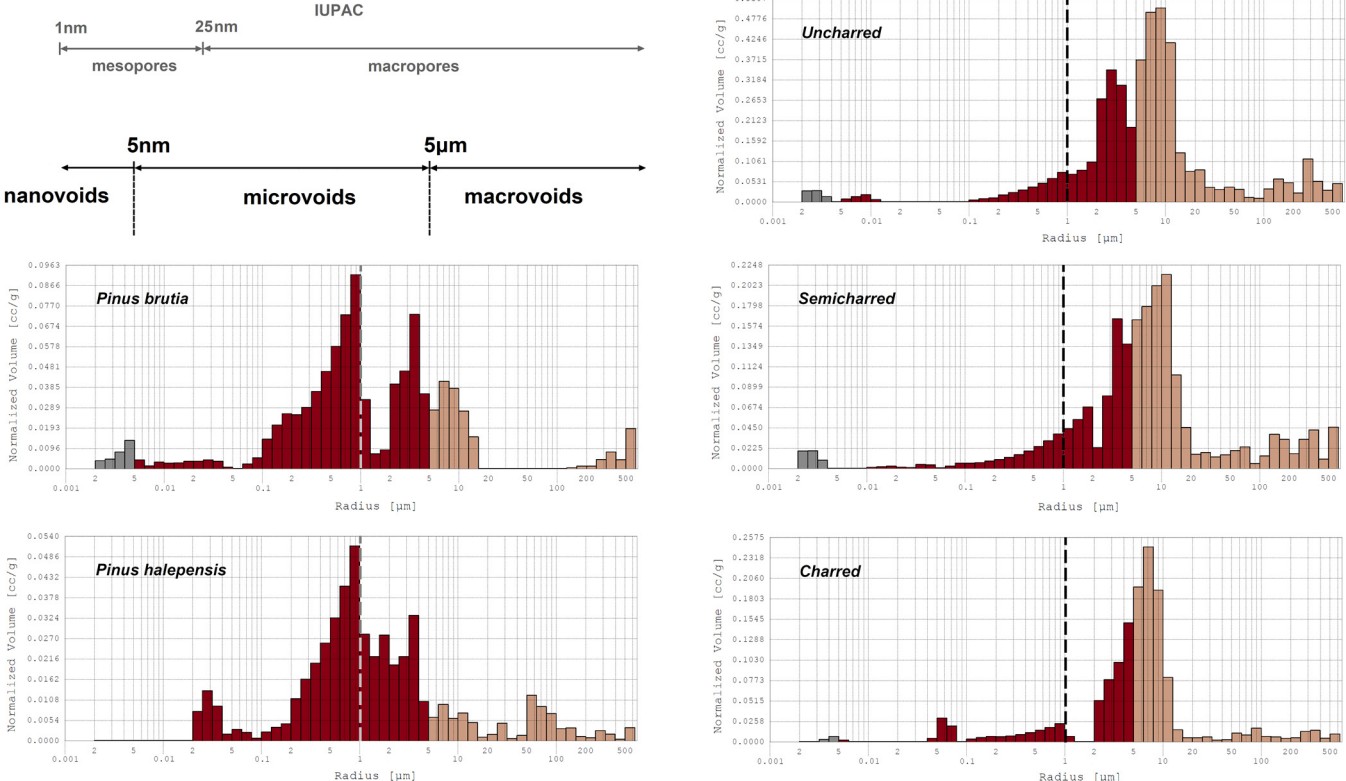

**Figure 5.** Histograms of intruded pore volume as a function of the pore radius of controls (*P. brutia* and *P. halepensis*) and archaeological wood (uncharred, semi-charred and charred).

Lastly, uncharred material presented the highest porosity recorded. As mentioned earlier, this was principally due to biodeterioration, as evident from SEM examination. Furthermore, its porosity was slightly higher than the semi-charred material and much higher in comparison to the charred wood, which had the smallest number of large pores, located mostly at the narrow range of 5 to 15 μm.

Porosity differences among the three charring conditions were also demonstrated with the pore size distribution curves as a function of the intruded volume (Figure 6). In these curves it became clear that the porosity of all three charring conditions increased compared to the reference samples of *P. halepensis* (0.50 cc/g) and *P. brutia* (0.92 cc/g). Furthermore, differences among the three charring conditions were again evident as the uncharred wood showed the highest intruded Hg volume (4.45 cc/g) compared to the semi-charred (2.05 cc/g) and the charred wood (1.40 cc/g).

The porosity of all three charring conditions, as anticipated, was inversely correlated with the basic density values recorded, as the denser a material, the less likely there will be voids present within.

Based on these results, and the fact that porosity greatly affects the impregnation rate [98] and the polymer retention during wood treatments, especially inside pores with diameters > 0.1 μm [102], it is expected that uncharred archaeological wood will be permeable and thus it will promote diffusion and allow a successful consolidation. In contrast, it is considered that the charred material will be resistant to diffusion even by low molecular weight polymers.

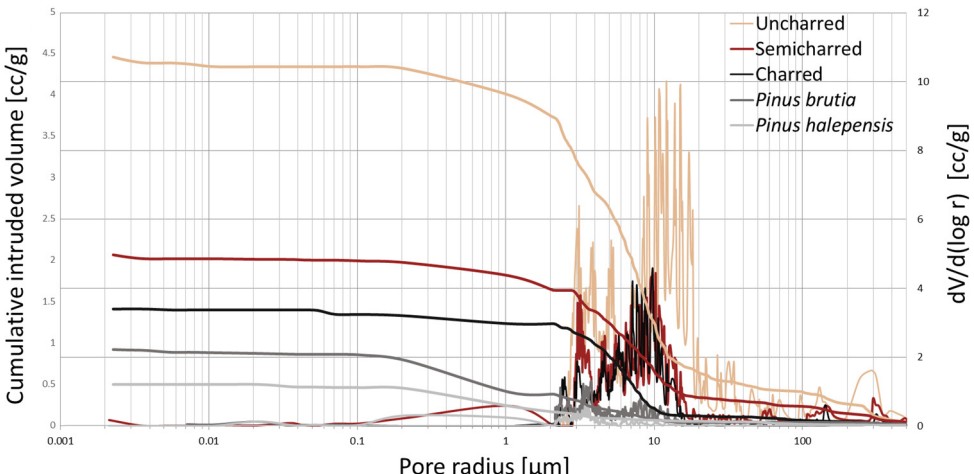

**Figure 6.** Differential pore-size distribution curves and cumulative intrusion curves of controls (*P. brutia* and *P. halepensis*) and the three conditions of the archaeological wood (uncharred, semi-charred and charred).

### 3.3. Mechanical Properties

3.3.1. Janka Hardness Test

Results obtained by employing the modified Janka test on the shipwreck material showed that the hardness of archaeological wood was considerably lower compared to the reference samples of *P. halepensis* and *P. brutia* (Figure 7, Table 2). This was rather justified as low hardness has been reported for deteriorated waterlogged archaeological wood [66] and because thermal exposure can also reduce hardness [24,27,73,77,86–88,103].

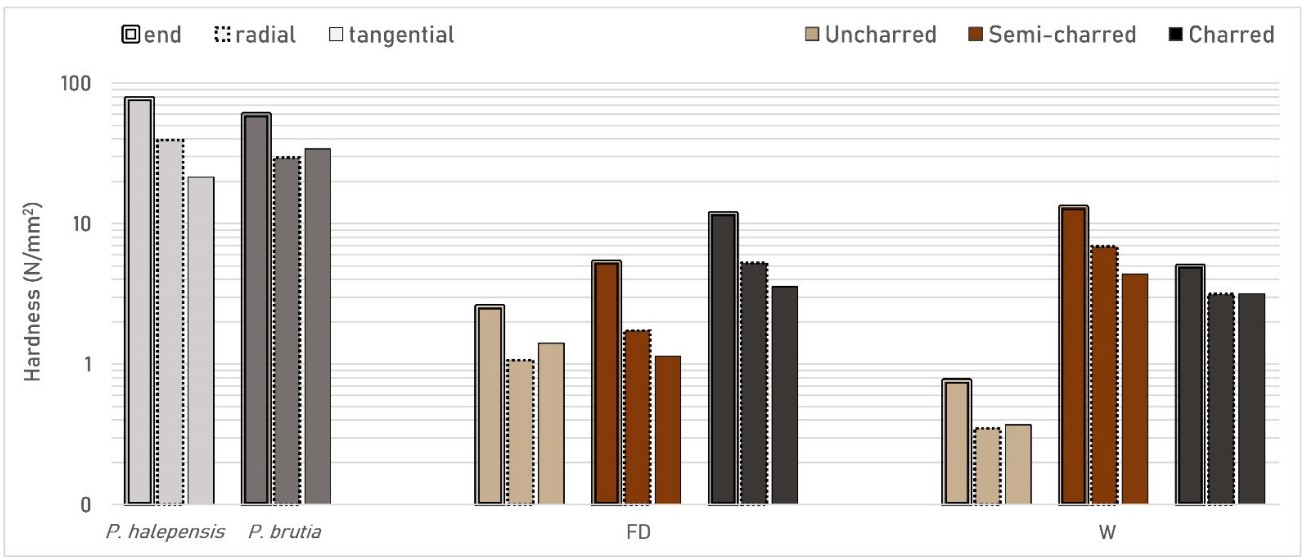

**Figure 7.** End and side (radial and tangential) hardness of the archaeological wood at the freeze-dried (FD) and at the waterlogged (W) state in comparison to the hardness of sound *P. brutia* and *P. halepensis,* of which earlywood-latewood values were averaged.

**Table 2.** Hardness values obtained with the modified Janka test. Values for sound *P. brutia* and *P. halepensis* are the average of four replicates. For the archaeological material, two replicates were used (* a single value due to material's failure). Values in brackets represent standard deviation.

| Hardness (N/mm$^2$) | | End | Radial | Tangential |
|---|---|---|---|---|
| *Pinus halepensis* | | 77.68 ($\pm$21.02) | 39.33 ($\pm$17.10) | 21.43 ($\pm$6.59) |
| *Pinus brutia* | | 59.76 ($\pm$26.02) | 29.53 ($\pm$16.32) | 34.09 ($\pm$16.00) |
| Uncharred | FD | 2.57 ($\pm$0.87) | 1.07 ($\pm$0.26) | 1.41 ($\pm$0.07) |
| | W | 0.76 ($\pm$0.18) | 0.35 ($\pm$0.02) | 0.37 ($\pm$0.00) |
| Semi-charred | FD | 5.32 ($\pm$1.02) | 1.73 ($\pm$0.13) | 1.14 ($\pm$0.20) |
| | W | 13.06 ($\pm$3.18) | 6.86 ($\pm$4.09) | 4.37 ($\pm$0.04) |
| Charred | FD | 11.76 * | 5.25 ($\pm$1.44) | 3.55 ($\pm$0.01) |
| | W | 4.98 ($\pm$3.46) | 3.16 ($\pm$1.15) | 3.17 ($\pm$1.07) |

The hardness values obtained among the three charring conditions varied as well (Figure 7). At the freeze-dried state, charred wood presented the highest hardness value, uncharred demonstrated the lowest and the semi-charred demonstrated an intermediate hardness. In contrast, at the waterlogged state, the hardness of the semi-charred material was the highest among the three charring conditions, followed by the charred and the uncharred material, which was again the lowest recorded.

The higher hardness of the semi-charred material compared to the charred could be also attributed to the different exposure of the materials to heat (temperature and duration), which may affect greatly the mechanical properties of the wood [73,88]. Wood subjected to thermal modification at temperatures from 180 to 250 °C can demonstrate lower hardness than unheated wood [24,27,73,77,88], and as above 250 °C the material is thermally degraded rather than modified [60], wood exposed to temperatures above 300 °C also presents a conspicuous decrease in hardness [86,87,103].

The hardness values recorded for the semi-charred material deviated between the two states (dry and waterlogged). This can be attributed to the transitional nature of the semi-charred material itself. The resolution of the hardness test applied (~1 mm indentation) could demonstrate small hardness differences in respect to its thermal degradation, and thus, values varied depending on how far the area tested was from the fire front. Thus, it is quite possible that for the freeze-dried sample, measurements were taken towards the uncharred decayed zone, whereas in the waterlogged sample towards the charred zone.

Lastly it should be noted that the high standard deviation values recorded for the reference samples (Table 2) demonstrate differences between earlywood (EW) and latewood (LW) [104–106].

### 3.3.2. Penetrometer

Results obtained with the fruit penetrometer for the three charring conditions are graphically presented in Figure 8. Uncharred wood demonstrated the lowest resistance to penetration (430–699 g), semi-charred wood presented almost a double increase in resistance (903–1156 g), whereas charred wood demonstrated the highest resistance (1147–1452 g). However, it should be noted that although a full-length penetration (2 cm) was attempted, the final penetration depth was not the same for all charring conditions. Uncharred wood allowed a 2 cm depth penetration, the semi-charred a partial penetration (~0.5–1 cm), whereas the charred wood allowed only a superficial penetration (~0–0.3 cm). Hence, results cannot be interpreted as hardness values as in hardness tests either the force [28] or the penetration depth [21,28] must be kept constant.

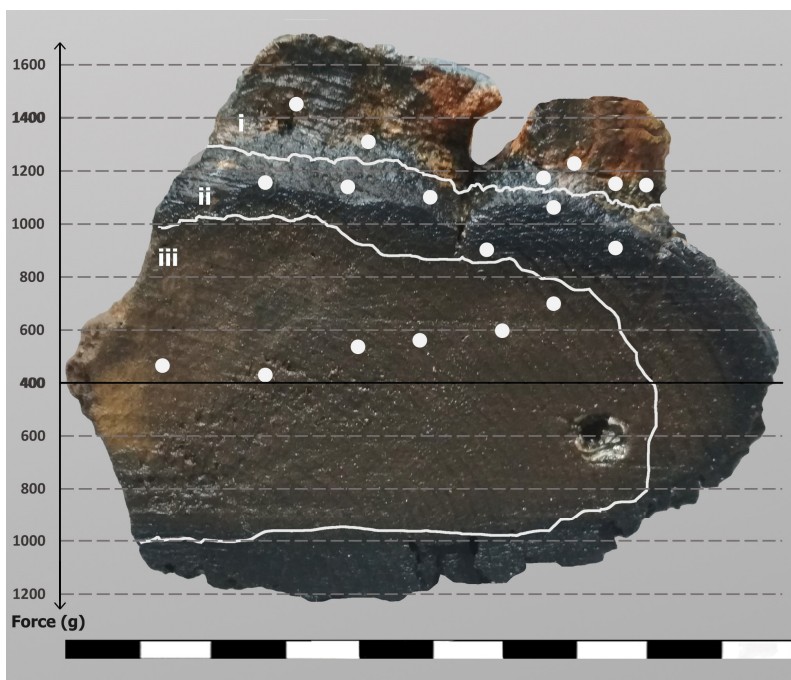

**Figure 8.** Graphical representation of the resistance to penetration values recorded on a sample, where all three charring conditions charred (i), semi-charred (ii) and uncharred (iii) coexisted.

Nonetheless, the results obtained demonstrated clearly the three distinct charring conditions, indicating that the fruit penetrometer, if properly calibrated, could be developed as a portable minimally invasive tool for identifying the existence of different charring conditions among the timbers.

Furthermore, among the three charring conditions, resistance to penetration measurements showed a correlation with the Rg values (Figure 9). Correlation between Rg and resistance to penetration has been reported for waterlogged archaeological wood [40,107], and is also anticipated to exist for semi-charred and charred material. However, further research is required to confirm this assumption.

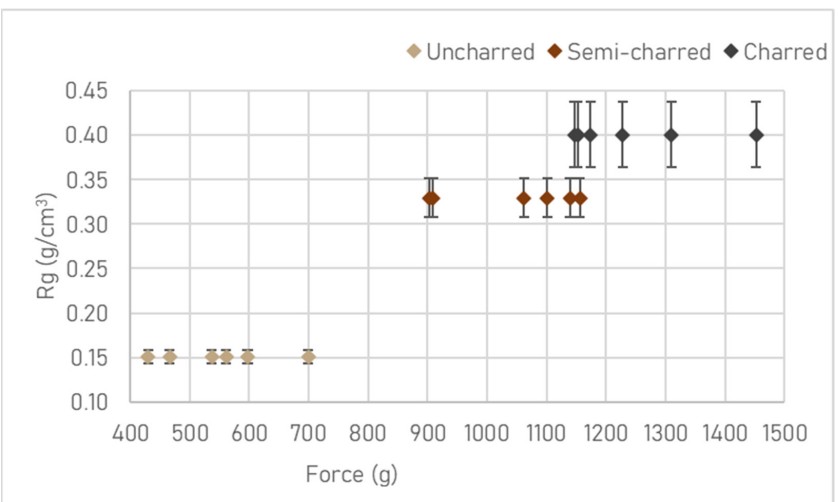

**Figure 9.** Correlation between resistance to penetration and Rg for the three charring conditions of uncharred, semi-charred and charred archaeological wood. Penetration points in each charring condition are plotted against the average Rg values of each condition calculated using four replicates. The error bars represent the standard deviation of Rg.

## 4. Conclusions

This study demonstrated that the physico-mechanical properties of the Rhodes shipwreck timbers varied greatly among the three charring conditions, and they were correlated to the residual chemistry of the wood.

The uncharred material showed typical physical properties of severely deteriorated waterlogged wood. It presented the highest moisture content and shrinkage and the lowest density among the other conditions. Its low hardness and resistance to penetration along with its increased porosity indicates that remedial conservation with a consolidation agent is required.

Charred wood, in contrast, presented the lowest moisture content and shrinkage and the highest density and resistance to penetration among the three charring conditions. It also showed the lowest porosity, suggesting low permeability and thus resistance to consolidation via diffusion. Nonetheless, based on the negligible shrinkage values, it is anticipated that the material may be safely air-dried without treatment.

The semi-charred wood was shown to be a transitional zone between the charred and the uncharred material, and thus, it predictably demonstrated intermediate values for almost all physicomechanical properties investigated in respect to uncharred and charred material. Regarding its conservation requirements, it is possible that the material can be air-dried untreated, nonetheless further research is considered necessary due its transitional nature.

Lastly, it became apparent that the most problematic timbers of the shipwreck are those where all three charring conditions coexist. For these timbers, the consolidation of the inner uncharred core will be rather problematic as the outer charred layer is expected to restrain the diffusion of even low molecular weight consolidants.

**Author Contributions:** Conceptualization, A.P.; methodology, A.P. and E.M.; investigation, E.M.; data curation, A.P., E.M. and N.-A.S.; writing—original draft preparation, A.P. and E.M.; writing—review and editing, A.P. and E.M.; supervision, A.P. All authors have read and agreed to the published version of the manuscript.

**Funding:** This research received no external funding.

**Institutional Review Board Statement:** Not applicable.

**Data Availability Statement:** Not applicable.

**Acknowledgments:** The authors would like to thank G. Koutsouflakis, for providing the archaeological material and A. Karampotsos for his technical support with the SEM.

**Conflicts of Interest:** The authors declare no conflict of interest.

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
