# Peer review of "Physico-Mechanical Properties of Waterlogged Archaeological Wood: The Case of a Charred Medieval Shipwreck"

_forests, doi:10.3390/f14030560_

Round 1
Reviewer 1 Report
The presented study is highly interesting. The manuscript is very well written and easy to understand. However, some polishing of the English language may be needed.
I recommend publication of the manuscript after considering the following comments:
Abstract: Needs to get shortened by 50 words (see journal’s requirements)
L58/59: Please rephrase: unclear what the ‘properties’ of […] ‘conditions’ are.
L60: ‘the term preservation’ is somewhat misleading since it refers to the ‘impregnation of wood with biocidal wood preservatives’.
Section 2: Photos of timbers representing the three different states (uncharred, semi-charred, charred) would be helpful to understand the following.
Section 2.1: The number of replicate samples should be provided for every test. It should also be indicated from where exactly the samples have been taken.
Section 2.2: What was the reason for not determining the oven-dry density of the samples? This would have allowed a comparison with finding from other previous studies.
L92 and elsewhere: Preferably the term ‘specimens’ should be used instead of ‘samples’.
L96: What was the reason for using 21°C/45%RH instead of the normal climate (i.e. 20°/65%RH)?
Eqs. 1 ff: weight or mass is usually abbreviated with ‘m’. Furthermore, not all of the abbreviations are explained in the respective legends.
Section 2.3.1, para 1: Actually, this is ‘discussion’, and not ‘Materials and methods’.
Section 2.3.1: What has been the reason to examine ‘hardness’? The more critical properties would have been impact bending strength or MOE. Others, for instance, looked into the brittleness of water-logged wood, e.g.:
Rapp, A.O., Brischke, C., Welzbacher, C.R., Nilsson, T., Björdal, C. 2008: Mechanical strength of wood from the Vasa shipwreck. Stockholm: The International Research Group on Wood Protection, IRG/WP/08-20381.
L 131 and elsewhere: Botanical genus names should be abbreviated after first time mentioned in the text.
L136 and elsewhere: More common would be: 20 x 20 x 20 (tang. X rad. long.) mm³
Section 2.3.2: Again, it would be interesting to know why ‘pin-tests’ were selected. More often drilling resistance measurements were applied to determine the impact of decay.
Heading 3.1: ‘Anatomical characteristics’ would be more adequate, since ‘SEM’ is a method, not a result.
L193-195: Which fungi are able to degrade wood at water-saturation? Please discuss. See also L.210.
L 243: The term ‘fiber saturation point’ is misleading. Neither fibers are saturated nor is it an exact point. More appropriate would be ‘cell wall saturation’. See for instance:
Thybring, E. E., Fredriksson, M., Zelinka, S. L., & Glass, S. V. (2022). Water in wood: a review of current understanding and knowledge gaps. Forests, 13(12), 2051.
Table 1 and Fig. 3 as well as Table 2 and Fig. 4: The same results are displayed twice. Such double presentation shall be avoided. Please delete one or the other.
L264: ‘Glass and Zelinka 2010’ not cited correctly, see journal’s requirements.
Table 1: Please provide the same number of decimals.
Table 1 and Fig. 3: What has been the SD? Have any statistical significance tests performed?
Figure 3 and 4: Botanical names please in italics.
Figure 4: SD? Are the 6 left columns recent materials?
Figure 7. These figures are way too small and thus unreadable. Furthermore, black font shall be used.
L369: Statement not fully correct: quite some papers address the performance/durability of TMT in marine environment, e.g.
Westin, M., Rapp, A., & Nilsson, T. (2006). Field test of resistance of modified wood to marine borers. Wood Material Science and Engineering, 1(1), 34-38.
Janus, M., Cragg, S., Brischke, C., Meyer-Veltrup, L., & Wehsener, J. (2018). Laboratory screening of thermo-mechanically densified and thermally modified timbers for resistance to the marine borer Limnoria quadripunctata. European Journal of Wood and Wood Products, 76, 393-396.
Klüppel, A., Cragg, S. M., Militz, H., & Mai, C. (2015). Resistance of modified wood to marine borers. International Biodeterioration & Biodegradation, 104, 8-14.
Ormondroyd, G., Spear, M., & Curling, S. (2015). Modified wood: review of efficacy and service life testing. Proceedings of the Institution of Civil Engineers-Construction Materials, 168(4), 187-203.
Author Response
Dear Sir/Madam
We thank you sincerely for your time and your valuable comments.
We have corrected the errors and incorporated almost all your suggestions and amendments. We feel that the paper after your contribution has been really improved.
You may find the point by point responses to your questions in the attached .pdf file

Reviewer 2 Report
In this paper, the authors report the physico-mechanical properties of three degrees of charring waterlogged archaeological wood. In general, the manuscript addresses an interesting topic. However, I would like to point some specific comments below:
1. Line 12-13. The major chemical differences of wood were not investigated in the whole manuscript, so the description of the abstract should be corrected.
2. Line 29-30. The number of keywoods in the manuscript should be reduced.
3. Table 1. The error ranges of EMC, MC of archaeological wood and reference wood are proposed to be added.
4. Line 402 and line 406. The question is why the uncharred waterlogged material and charred wood both presented the lowest moisture content
It is a very interesting study. And, it is suggested that the analysis of the chemical properties of waterlogged archaeological wood should be explored in future work.
Author Response
Dear Sir/Madam
We thank you very much for your comments. We have incorporated them all in the manuscript and we believe that they have really improved the paper.
Below you may find the point by point answers to your questions.
- 1. Line 12-13. The major chemical differences of wood were not investigated in the whole manuscript, so the description of the abstract should be corrected.
The abstract has been corrected in order not make clear that the chemical properties of the archaeological material have been investigated elsewhere.
- Line 29-30. The number of keywords in the manuscript should be reduced.
The number of keywords is reduced
- Table 1. The error ranges of EMC, MC of archaeological wood and reference wood are proposed to be added.
Standard deviation values have been added in the table of physical properties examined
Reference wood SD values, as obtained by the literature, were not available.
- Line 402 and line 406. The question is why the uncharred waterlogged material and charred wood both presented the lowest moisture content
This was a copy-paste error. The sentence has been revised and it now agrees with results presented earlier in the study (tables and graphs), where the uncharred waterlogged wood presented the highest MC and lowest density, whereas the charred wood had the lowest MC and the highest density.
- It is a very interesting study. And, it is suggested that the analysis of the chemical properties of waterlogged archaeological wood should be explored in future work.
Thank you very much for your comment. The chemical profile of the material has been investigated by the authors and published in Forests at https://www.mdpi.com/1999-4907/12/11/1594.
The introduction of the ms has been now modified in order to clarify this, and to explain that it was the chemistry of this waterlogged archaeological material that has triggered the present work.
Reviewer 3 Report
This paper reports on Physico-mechanical properties of waterlogged archaeological wood. However, there are several questions not mentioned or clearly clarified by the authors, so the manuscript should be revised before publication.
1)In line 78, page 2, please clearly indicate whether all three types of specimens are waterlogged
samples
2)In line 265, page 7,the symbol of density should be changed to g/cm3ï¼›Moreover, the symbol of shrinkage should be consistent with Table 1 and should be changed to βcross.
3)In line 294, page 8, it is mentioned “In contrast at the waterlogged (W) state, the hardness of the semi-charred material was the highest among the three charring conditions and followed by the charred one, whereas the hardness of the uncharred material was the lowest recorded. ”. However, the specific reason for the highest hardness of semi-charred when at the waterlogged state, is not clearly indicated.
4) In line 402, page 12, it is mentioned that "It presented the lowest moisture content and density", but it is clearly stated that uncharred waterlogged material has the highest moisture content, please correct.
Author Response
Dear Sir/Madam
We thank you very much for your comments. We have incorporated them in the manuscript and we feel that the paper has been really improved.
Below you may find the point by point answers to your comments
- In line 78, page 2, please clearly indicate whether all three types of specimens are waterlogged samples
All samples were waterlogged. The sentence is revised to better clarify this
- In line 265, page 7,the symbol of density should be changed to g/cm3ï¼›Moreover, the symbol of shrinkage should be consistent with Table 1 and should be changed to βcross.
The density symbol is changed to g/cm3 and the symbol of shrinkage in now consistent
- In line 294, page 8, it is mentioned “In contrast at the waterlogged (W) state, the hardness of the semi-charred material was the highest among the three charring conditions and followed by the charred one, whereas the hardness of the uncharred material was the lowest recorded. ”. However, the specific reason for the highest hardness of semi-charred when at the waterlogged state, is not clearly indicated.
A sentence has been added in order to provide an hypothesis that justifies the highest hardness of the semi-charred material at the waterlogged state.
- In line 402, page 12, it is mentioned that "It presented the lowest moisture content and density", but it is clearly stated that uncharred waterlogged material has the highest moisture content, please correct.
This was a copy-paste error. The sentence has been revised and it now agrees with results presented earlier in the study (tables and graphs), where the uncharred waterlogged wood presented the highest MC and lowest density, whereas the charred wood had the lowest MC and the highest density.
Reviewer 4 Report
The article entitled „Physico-mechanical properties of waterlogged archaeological wood: The case of a charred medieval shipwreck” presents a thorough characterisation of differentiated waterlogged wooden material from an ancient shipwreck. The experiments were well-planned and conducted, and the results are presented clearly and described in detail. The results obtained are useful from the conservation perspective and will be helpful for researchers and wood conservators dealing with partly or unevenly charred wooden artefacts.
Suggestions for some small editorial changes can be found in the attached file.

Author Response
Dear Sir/Madam
We thank you very much for your comments and suggestions.
The editorial changes suggested have been incorporated in the manuscript and we feel that they have really improved the paper.